# Real-Time 3D Object Detection on Crowded Pedestrians

**DOI:** 10.3390/s23218725

**Published:** 2023-10-26

**Authors:** Bin Lu, Qing Li, Yanju Liang

**Affiliations:** 1Institute of Microelectronics of the Chinese Academy of Sciences, Beijing 100029, China; lubin@wiot.tech (B.L.); liangyanju@wiot.tech (Y.L.); 2University of Chinese Academy of Sciences, Beijing 101408, China; 3Wuxi IoT Innovation Center Co., Ltd., Wuxi 214028, China

**Keywords:** attention, center alignment, heatmap, point sampling

## Abstract

In the field of autonomous driving, object detection under point clouds is indispensable for environmental perception. In order to achieve the goal of reducing blind spots in perception, many autonomous driving schemes have added low-cost blind-filling LiDAR on the side of the vehicle. Unlike point cloud target detection based on high-performance LiDAR, the blind-filling LiDARs have low vertical angular resolution and are mounted on the side of the vehicle, resulting in easily mixed point clouds of pedestrian targets in close proximity to each other. These characteristics are harmful for target detection. Currently, many research works focus on target detection under high-density LiDAR. These methods cannot effectively deal with the high sparsity of the point clouds, and the recall and detection accuracy of crowded pedestrian targets tend to be low. To overcome these problems, we propose a real-time detection model for crowded pedestrian targets, namely RTCP. To improve computational efficiency, we utilize an attention-based point sampling method to reduce the redundancy of the point clouds, then we obtain new feature tensors by the quantization of the point cloud space and neighborhood fusion in polar coordinates. In order to make it easier for the model to focus on the center position of the target, we propose an object alignment attention module (OAA) for position alignment, and we utilize an additional branch of the targets’ location occupied heatmap to guide the training of the OAA module. These methods improve the model’s robustness against the occlusion of crowded pedestrian targets. Finally, we evaluate the detector on KITTI, JRDB, and our own blind-filling LiDAR dataset, and our algorithm achieved the best trade-off of detection accuracy against runtime efficiency.

## 1. Introduction

In the field of autonomous driving, pedestrian detection under 3D point clouds is an indispensable part of the whole autonomous driving system. Because the 3D point cloud has rich depth information, it can provide accurate distance information for automatic driving, thus providing a reliable judgment basis for the path-planning system. Early research, based on a single image to detect pedestrians, showed that it is difficult to recover the depth information. Although this was later developed to use binocular cameras or depth cameras to obtain depth information, the depth information obtained by this method is still not accurate enough, or even wrong. With the widespread use of LiDAR in the field of automatic driving, LiDAR-based pedestrian detection began to develop and is widely used. With the improvement of LiDAR resolution and the development of pedestrian-detection methods, LiDAR-based 3D point cloud detection has gradually become a key component of autonomous driving applications. However, 3D point clouds have sparse and disordered characteristics, and 3D point clouds cannot provide dense texture features like images. The primary concern for pedestrian detection in 3D point clouds is to overcome these problems. Many studies have meaningfully explored 3D point cloud detection and achieved high detection accuracy in some scenarios. However, the 3D detection of pedestrians under blind-filling LiDAR is still a challenging task for two main reasons: first, the laser beam of blind-filling LiDAR is sparser and has lower vertical angular resolution, resulting in only a few laser reflection points for objects farther away from the LiDAR, and some conventional point-sampling methods or voxel-quantization methods may increase the risk of losing these critical points. Second, since the blind-fillingLiDAR is typically mounted on the side, when pedestrians gather in close proximity to the LiDAR, occlusion can significantly reduce the recall of pedestrian detection.

In this paper, we design three modules for solving the problem of low recall under occlusion while keeping the time consumption low. Although point cloud data are more sparse compared to images, they still have significant redundancy. To further speed up the detection of the model, it is necessary to remove redundancy from the point cloud. However, the traditional radial-distance-based farthest-point-sampling (DFPS) [1] method leads to the omission of critical points. In order to further consider the clustering features of the point cloud, we design a more-efficient point-cloud-sampling method, which not only effectively reduces the point cloud redundancy, but also fully considers the feature information of the key points. The point cloud formed by LiDAR is distributed on the surface of the object and will be concentrated on the side of the target facing the LiDAR. This special nature may lead to a misalignment in the regression of the center position of the target. To mitigate the problem of center alignment failure, we use quantization in the polar coordinate system of the point cloud space, and the quantized mesh will show a dispersion distribution from near to far according to the distance from the LiDAR. Along the radial direction, when the target appears in the current grid, the neighboring grids are likely to lose the target keypoints due to occlusion. In this case, we simply use the sharing of adjacent grid features to compensate for the unaligned target center error due to its own occlusion. In a 3D point cloud, the mixing between the background and target occupancy points reduces the detection accuracy. We find that this is mainly due to the fact that the regular convolution kernel on the convolutional layer does not adapt to the rich object contours when extracting features from the point cloud, resulting in rough localization. Therefore, we design an attention mechanism that can adapt itself to this contour variation and use a heatmap of the target center location to guide the training of this attention module to achieve higher detection accuracy.

Combining the aforementioned modules, we present a real-time crowded pedestrian (RTCP) detector, which can be trained end-to-end and provides real-time 3D detection of pedestrian targets. We evaluate the performance on KITTI [2], JRDB [3], and our own dataset. Compared with the counterparts, RTCP achieved the best balance between effectiveness and real-time performance. Our contribution points are as follows:We propose a point-cloud-sampling method based on clustering features, which reduces redundancy while effectively reducing the risk of target key point missing.We introduce a point-cloud-spatial-quantization method based on polar coordinates and use adjacent grid feature sharing to improve the accuracy of the target center anchor position.We use an attention mechanism to refine the receptive field and use the target location heatmap to guide training, so as to obtain higher detection accuracy.

## 2. Related Works

### 2.1. Point Cloud Representation

The structural characteristics of 3D point clouds are different from those of images, where the image is dense, while point clouds are unordered and sparse. It is difficult to directly obtain useful information from point clouds. In order to obtain structured features, it is necessary to carefully design strategies for extracting point cloud features.

Voxnet [4], VoxelNet [5], Octnet [6], and SECOND [7] organize point clouds by voxelization. They quantize the 3D space occupied by a certain range of point clouds to form many regularly arranged small cubes and merge the features of the points belonging to each cube according to their coordinate positions. The merged feature vectors are the features of all the points in that small cube, and finally, these vectors are arranged according to the quantized positional relationships to form a feature tensor. Although this processing method is effective, it also introduces significant quantization errors, and the detection accuracy is directly related to the quantization granularity. In order to speed up the preprocessing of the point cloud and avoid using time-consuming 3D convolution for the 3D tensor, PointPillar [8] merges the features of all the small cubes along the direction of one of the XYZ coordinate axes and achieves a good balance between detection accuracy and time consumption. Obtaining clustered features directly on the point cloud will improve the recall of the target compared to point cloud quantization. PointNet [9] proposes a method to extract features directly from disordered and sparse point clouds using linear layers and symmetric functions. However, the direct extraction of features on the point cloud is inefficient, and the computational complexity will increase exponentially as the density of the point cloud increases. In order to reduce the redundancy of the point cloud, PointNet++ [1], IA-SSD [10], and 3DSSD [11] propose different point-cloud-sampling methods. The point-cloud-sampling methods not only reduce the number of points in the point cloud, but also retain more-effective point clouds, thus achieving a balance between time consumption and effectiveness. Due to the sparsity of the point cloud itself, there are many zero feature vectors in its quantized feature tensor. It is inefficient to extract features on this feature tensor directly using 3D convolution, in which case features can be extracted using spConv [12]. It is worth mentioning that, in many studies, the previously mentioned methods can be used in combination. In the voxelization approach, for points within the same mesh, the method proposed by PointNet [9] is also used to obtain local features of the mesh. Prior to the adoption of spConv [12], some networks would use the point sampling method proposed by PointNet++ [1] to reduce the number of points in the point cloud in order to speed up the computation.

### 2.2. Three-Dimensional Point Cloud Detection

With the increasing popularity of LiDAR applications, 3D target detection based on point clouds has also been rapidly developed. We can usually classify them into two categories based on the pipeline of detection models: single-stage detection and two-stage detection. Similar to image-based target detection, two-stage 3D point-cloud-based target detection, as in [13,14,15], focuses on dealing with the foreground and background issues in the first stage and uses a region-proposal network (RPN) to narrow down the search range of the target, and in the second stage, it refines the center position and bounding box of the object in the ROI. Although two-stage methods can achieve higher detection accuracy, they operate at a slower speed. Single-stage detection balances detection accuracy and time consumption, and these methods focus more on how to extract information related to the target from the point cloud. Simony et al. [16] extracted point cloud features from a bird’s eye-view (BEV), and Wu et al. [17] obtained the feature tensor in the front-view panorama (FOV). Li et al. [18] used voxel quantization and then extracted features using 3D convolution. Lang et al. [8] stacked voxels on the Z-axis to form columnar features, while Yan et al. [7] used an improved sparsity strategy to optimize 3D convolution. In addition, in many scenarios, there is a need to balance detection accuracy and time consumption, and many studies have made meaningful attempts. PIXOR [19] encodes the point cloud in the bird’s eye-view to reduce the complexity of encoding the point cloud, and to further improve the efficiency of the model, PIXOR uses a lightweight backbone and a simplified header. The methods 3DSSD [11] and CIA-SD [20] use point sampling to reduce the computational complexity. TANet [21] and PiFeNet [22] add an attention mechanism when extracting features using a convolutional network, and high detection accuracy can be achieved on a lightweight backbone network. Inspired by these studies, we propose a lightweight end-to-end real-time pedestrian detector.

## 3. Proposed Framework

### 3.1. Point Cloud Sampling

Inspired by PointNet++ [1], IA-SSD [10], 3DSSD [11], and SASA [23], we use point sampling to preprocess the point clouds. Using point sampling can reduce the redundancy of point clouds and, thus, reduce the computational complexity of subsequent modules. The basic premise of the point sampling method is to ensure sufficient effective point clouds and eliminate redundant background point clouds. However, due to the lack of prior knowledge of the target’s position, it is almost impossible to fully preserve the points on the target. Usually, we use farthest point sampling (FPS), which treats all points equally and can maximize the preservation of the distribution characteristics of the point cloud.

On many datasets, we have found through statistical results that point cloud data themselves can also reduce unnecessary redundant data through some height priors, such as ground and sky point clouds. We separately calculated the positions of the upper and lower vertices of pedestrians on KITTI and our own dataset, as shown in Figure 1. On the KITTI dataset, the point clouds of pedestrians are mainly concentrated between [−3,1], while our dataset is concentrated between [−2.5, 0.5]. Before sampling the entire data, we can filter some points through the positions on the Z-axis, as shown in Figure 2. On our dataset, before and after filtering, the number of invalid points in the ground and sky decreased significantly.

After undergoing distance-based FPS (DFPS) filtering, we still need to continue reducing the number of point clouds to improve the computational efficiency. We used DFPS as a coarse point-cloud-filtering step. Many research works have proposed some effective resampling strategies for point clouds. For example, IA-SSD [10] uses multiple MLP layers to obtain scores for each point and then filters the TopK points based on the scores. The method 3DSSD [11] uses the similarity between feature vectors as the metric of FPS, taking more semantic information into consideration.

Inspired by these strategies above, we propose an attention point-resampling (APR) module, as shown in Figure 3. First, in the bird’s eye view, we quantize the point cloud space into small regularly arranged cubes. The center of the cube 
(i,j)
 after quantization is 
$\left(X_{i},Y_{j}\right)$
. All the cubes are regularly arranged in the X-Y plane and, in the overhead view, form a plane F. The size of F is 
W×H
. We used an MLP layer with shared weights for all points to obtain new features for each point. Then, the features of these points are projected back onto F according to their locations, and when multiple points are projected into the same cube, we use the feature vectors of the summed multiple points to obtain the fused feature vectors 
$v_{ij}\in R^{C}$
 and fill in the cubes where no points fall with the zero vector 
$0\in R^{C}$
. After these steps, a tensor feature 
$F^{\prime }$
 of 
$R^{W\times H\times C}$
 is obtained.

### 3.2. Quantization in Polar Coordinates

In order to solve the occlusion problem of the object itself, we quantized the point cloud space according to the radial direction of the laser divergence direction of LiDAR, which is more in line with the structural characteristics of the point cloud. We adopted the polar coordinate quantization strategy proposed by Manoj Bhat et al. [24] and Yang Zhang et al. [25]. It is very simple to convert from Cartesian coordinates 
$\left(x_{i},y_{i}\right)$
 to polar coordinates 
$\left(\phi_{i},r_{i}\right)$
, As shown in Formula (Equation 1):
(1)
$$\begin{array}{c} \phi_{i}=arctan\left(\frac{y_{i}}{x_{i}}\right),\phi_{q}=\left\lfloor \frac{\phi_{i}}{\phi_{step}}\right\rfloor ,\phi_{step}=\frac{2\pi }{H}\\ r_{i}=\sqrt{x_{i}^{2}+y_{i}^{2}},r_{q}=\left\lfloor \frac{r_{i}}{r_{step}}\right\rfloor ,r_{step}=\frac{r_{max}}{W} \end{array}$$


After converting the Cartesian coordinates 
$\left(x_{i},y_{j}\right)$
 to polar coordinates 
$\left(\phi_{i},r_{i}\right)$
, where 
$\phi_{i}$
 is the angle, 
$r_{i}$
 is the radius, 
$\phi_{q}$
 is the index of the angular position after quantization and rounding upwards, 
$\phi_{step}$
 is the quantization step of the angle, 
$r_{q}$
 is the index in the radial directionobtained by quantizing in the radial directionand rounding upwards, and 
$r_{step}$
 is the quantization step of the radius, wequantify it into W equal-length intervals in the radial direction and H equal-angle intervals in the angular direction. As shown in Figure 4, after quantization, we only need to project the points onto the quantified plane according to their polar coordinates.

As shown in Figure 4, we project each point onto a quantized interval 
$S_{\left(\phi ,r\right)}$
 and make weighted summation of all 
(i,j)
 points’ feature vector 
$F_{\left(i,j\right)}$
 in 
$S_{\left(\phi ,r\right)}$
 to obtain the new feature vector 
$F_{\left(\phi ,r\right)}$
, where 
$w_{i}^{\phi }$
 is the weight on the angle and 
$w_{j}^{r}$
 is the weight in the radial direction:
(2)
$$\begin{array}{c} w_{i}^{\phi }=1.0-\left(\frac{\phi_{i}}{\phi_{step}}-\left\lfloor \frac{\phi_{i}}{\phi_{step}}\right\rfloor \right)\in \left[0,1.0\right]\\ w_{j}^{r}=1.0-\left(\frac{r_{j}}{r_{step}}-\left\lfloor \frac{r_{j}}{r_{step}}\right\rfloor \right)\in \left[0,1.0\right]\\ F_{\left(x,y\right)}=\sum_{\left(i,j\right)\in S_{\left(\phi ,r\right)}}w_{i}^{\phi }w_{j}^{r}F_{\left(i,j\right)} \end{array}$$


These 
W×H
 newly obtained feature vectors 
$F_{\left(i,j\right)}\in R^{1\times C},i\in \left\{1,2,3,\cdots ,W\right\},$

j∈{1,2,3,⋯,H}
 together form a new feature tensor 
$F_{\left(\phi ,r\right)}\in R^{W\times H\times C}$
, where *C* is the dimension of the point feature vector.

We noticed that, in autonomous driving applications, the point cloud formed by laser projection on the surface of an object is often distributed on the surface facing the LiDAR side, which can have a negative impact on target detection. This issue was mentioned in [26], and a voting clustering method was used to obtain new anchor positions, effectively solving the problem. However, as the number of points increases, the computational complexity of the voting operation also increases. To improve the efficiency of the detector, we propose a simpler method.

According to the analysis of the point cloud and annotation results, we found that, if there is a target in the grid near the LiDAR in the polar coordinate system, its radially adjacent grid will lack key points due to the occlusion of the target itself. Therefore, as shown in Figure 4, we only need to fuse the features of radially adjacent grids in polar coordinates to compensate for the centroid alignment error caused by the target’s own occlusion.

### 3.3. Object Alignment Attention

Through the processing of the previous module, we obtained two types of feature maps, where 
$F_{1}\in R^{W_{1}\times H_{1}\times C_{1}}$
 is generated by the APR module and 
$F_{2}\in R^{W_{2}\times H_{2}\times C_{2}}$
 is generated by the CAP module. In experiments, we usually set 
$W_{1}=W_{2},H_{1}=H_{2}$
. Although both are based on the back-projection of the point positions to obtain the fused features of the points, the CAP module focuses more on the alignment relationship between the center of the quantized cube and the center of the target, whereas APR focuses on the semantic features of the original cloud of points during the point-resampling process. Therefore, directly connecting the two feature maps is sub-optimal, and they still need to be further aligned.


$F_{1}$
 and 
$F_{2}$
 are not fully aligned with the actual position of the target due to quantization errors. The best solution is to use the attention mechanism to further adjust the offset of the anchor points for target position prediction. PointNet [27] proposes a method to focus on the local structure of the point cloud so as to achieve the adjustment of the offsets, and the work in [28] proposes to improve the accuracy of predicting the target position by using the semantic information obtained from the graph attention convolution. Inspired by PCAN [29] and TANet [21], we propose a more-convenient attention method, called object alignment attention (OAA), as shown in Figure 5.

We use 
1×1
 convolutions on 
$F_{1}$
 and 
$F_{2}$
, respectively, to obtain 
$F_{1}^{\prime }\in R^{W\times H\times C}$
 and 
$F_{2}^{\prime }\in R^{W\times H\times C}$
. Then, we concatenate 
$F_{1}^{\prime }$
 and 
$F_{2}^{\prime }$
 to get feature tensor 
$f^{2\times W\times H\times C}$
.

(3)
$$\begin{array}{c} f^{2\times W\times H\times C}=concat\left(F_{1}^{\prime },F_{2}^{\prime }\right) \end{array}$$


Considering the effect of different pooling methods on positional alignment, we obtain two pooled features 
$F_{max}$
 and 
$F_{avg}$
 using two pooling methods for 
$f^{2\times W\times H\times C}$
.

As shown in Figure 5, the up branch, for 
$F_{1}^{\prime }$
 and 
$F_{2}^{\prime }$
 using a elementwise max pooling, 
$F_{max}$
 is obtained in 
$R^{W\times H\times C}$
, and we perform channelwise average pooling on 
$F_{max}$
 and, then, obtain a weight graph 
$w_{1}^{N\times W\times H}$
 through the sigmoid operation.

(4)
$$\begin{array}{c} F_{max}=\max_{N=2}\left(f^{2\times W\times H\times C}\right)\\ w_{1}=\mathrm{sigmoid}\left({\mathrm{mean}}_{C}\left(F_{max}\right)\right) \end{array}$$


In the down branch, we utilize an elementwise avg pooling on 
$f^{2\times W\times H\times C}$
 to obtain 
$F_{avg}$
. Next, the entire feature map is flattened to 
$f^{(W\times H)\times C}$
, and C number of vectors 
$v^{(W\times H)\times 1}$
 are passed through a shared weight MLP layer to obtain the weighted vector 
$w^{C\times 1}$
. As shown in Figure 5, finally, 
$w_{2}$
 is obtained through the weighting, sigmoid, and reshape operations.

(5)
$$\begin{array}{c} F_{avg}={\mathrm{mean}}_{N=2}\left(f^{2\times W\times H\times C}\right)\\ f^{(W\times H)\times C}=\mathrm{flatten}\left(F_{avg}\right)\\ w^{C\times 1}=w_{\mathrm{MLP}}\cdot f^{(W\times H)\times C}\\ w_{2}=\mathrm{reshape}\left\{\mathrm{sigmoid}\left\{f^{(W\times H)\times C}\cdot w^{\left(C\times 1\right)}\right\}\right\} \end{array}$$


We weight 
$F_{max}$
 and 
$F_{min}$
 with 
$w_{1}$
 and 
$w_{2}$
, respectively, to obtain 
$F_{1}^{o}$
 and 
$F_{2}^{o}$
. Finally, we perform an elementwise summation operation on 
$F_{1}^{o}$
 and 
$F_{2}^{o}$
 to obtain 
$F^{o}$
.

## 4. Framework

As shown in Figure 6, at the beginning of the network, we use DFPS and APR to sample the point clouds. This method preserves as many useful point clouds as possible while reducing the computational complexity of subsequent modules. Then, we use the CAP module to obtain the quantified feature tensor on the sampled point cloud. In order to align the predicted position of the feature tensors generated by APR and the CAP module in the X-Y plane with the position information of the target, we obtain the aligned features through the OAA module and utilize the labeled target-position-based heatmap to guide the training of the attention awareness layer. Finally, in order to ensure the invariance of target detection at multiple scales, we adopted a conventional FPN [30] structure to fuse features from multiple scales. Finally, we perform the classification prediction and boundary box regression on the fused feature tensor.

## 5. Experiments

We tested the detection performance of the proposed model on KITTI, JRDB, and our private dataset and compared it with several real-time models. In addition, we conducted ablation experiments on the APR, CAP, and OAA modules of the proposed model, respectively, used the normal point-cloud-sampling operation and voxelization method to replace them, and compared the effects of various module combinations.

### 5.1. KITTI and JRDB

We downsampled the point cloud along the z-axis for the KITTI dataset to make the point cloud data more sparse. Under this sampling, the number of target point clouds dropped, and some remote targets even had no key points. Therefore, we filtered out these labels without key points in the dataset. As shown in Figure 7, observing the downsampled point cloud data from the side, the number of key points occupied by pedestrian targets decreased. Through the experiments, we found that the detection effect on this point cloud significantly decreased. As shown in Table 1, we compared our detector with some effective real-time detectors on the KITTI validation set. Our model can achieve a running speed of over 50fps while ensuring the detection effect.

For the JRDB and KITTI datasets, our model utilizes two stages of the point-sampling operation: Stage 1’s sampling number was 4096, and the second stage’s sampling number was 2048. Their quantization parameters were slightly different. On the JRDB dataset, the APR module used 
X∈[−30,30]
, 
Y∈[−30,30]
, 
Z∈[−2,5]
 with quantization steps of 0.5, while the CAP module used 
R∈[0,30]
 with quantization steps of 0.25, angle 
ϕ∈[0,360]
 with quantization steps of 3. Adam was used as the gradient update strategy, with a smoothing constant of 
$\beta_{1}=0.9,\beta_{2}=0.99$
, with an initial learning rate of 100; the learning rate was updated once per epoch using cosine annealing, and a total of 45 epochs were trained. For the KITTI dataset’s parameter setting, refer to Section 6.

As shown in Table 1 and Table 2, we compared the performance of our model and common models with better real-time performance on the KITTI and JRDB validation sets. Our model can achieve a running speed of over 50fps while ensuring the detection effect.

### 5.2. Our Private Dataset

Our own blind-filling LiDAR was installed on the side of a truck at a height of 1.8 m, as shown in Figure 8. The blind-filling LiDAR mainly monitors pedestrians and vehicles within 10 m of the vehicle. We collected point cloud data from multiple scenarios, such as junctions, parking lots, and city roads, and finally obtained 5000 point cloud files, with a total of 83,067 labeled pedestrian targets. A large number of pedestrians in the dataset were concentrated from 3 to 8 m away from the vehicle, and pedestrians near the vehicle were crowded together, resulting in severe occlusion. As shown in Figure 9, we calculated the number of key points in the point cloud occupied by pedestrians within different distance intervals in the dataset. Due to the use of a 32-line LiDAR with a low longitudinal angle resolution, the point cloud occupied by the remote (usually 5 m away) pedestrian target was sparse. For targets beyond 10 m, as shown in the 3D box marked in green in Figure 8, we directly ignored it. Currently, we are unable to make the dataset publicly available due to commercial protection of the dataset, and we expect to open up access to the dataset in the next year.

We compared the performance of several detectors on this dataset, as shown in Table 3. Our model had the best detection performance for the dataset collected under the blind-filling LiDAR and can ensure real-time performance.

## 6. Ablation Studies

To further validate the effectiveness of our proposed detector, we conducted several sets of comparative ablation experiments on various parts of the RTCP network.

Our experiments were conducted on KITTI and our blind-filling LiDAR dataset, respectively. As the scenarios of the two datasets are completely different, there were some differences in our parameter settings. For the KITTI dataset, our first point sampling used DFPS with 4096 sampling points, and the second point sampling had 2048 sampling points. We selected a point range of 
X∈[−40,40]
, 
Y∈[0,72]
, 
Z∈[−3,1]
, with a quantization step of 0.4, using Adam as the gradient update strategy, with smoothing constant 
$\beta_{1}=0.9,\beta_{2}=0.99$
, and initial learning rate ob 
1e−2
, which was updated once per epoch using cosine annealing, and a total of 60 epochs trained. For the blind-filling LiDAR dataset, the sampling range was 
X∈[−15,15]
, 
Y∈[−15,15]
, 
Z∈[−2.5,0.5]
, with a sampling step of 0.15 and an initial learning rate of 
$1\times 10^{-3}$
.

In order to validate the effectiveness of the sampling module APR, we compared two point-cloud-sampling methods: DFPS+DFPS and DFPS+APR. When we used the DFPS+DFPS scheme, in order to allow it to interface with the OAA module, we quantized the point cloud produced by the second DFPS module and, then, fed the quantized results into the OAA module.

For the CAP module, it can be replaced by a regular voxel quantization method (as mentioned in PointPillar [8]). In the comparison experiments, the quantization parameters for CAP were set to radial distance 
R∈[0.15]
 with a step of 0.075, angle 
ϕ∈[−90,90]
 with a step of 0.9 on the blind-filling dataset, and on the KITTI dataset, using 
R∈[0.40]
 with a step of 0.2 and angle 
ϕ∈[−90,90]
 with a step of 1.0.

As mentioned earlier, after extracting the attention-based feature maps using the OAA module, we also introduced an additional auxiliary branch to guide the training, which predicts the target location heatmap. To demonstrate the effectiveness of this auxiliary branch, we compared the detection performance of the entire detector with and without this branch.

Finally, we summarize all the ablation experiments in Table 4. After the comparison, we found that:On the quantified feature map, it is harmful to directly utilize the heatmap branch to guide the training without using the CAP module;The CAP module improved the detector’s effectiveness more on the dataset of the blind-filling LiDAR;The attention module (CAP) guided by the heatmap branch can exert greater advantages and significantly improve the accuracy of the model.

## 7. Conclusions

Our proposed APR module can effectively reduce the redundancy of the point cloud while retaining more effective keypoints through semantic information; the CAP module employs polar quantization and neighborhood feature fusion to effectively solve the problem of the point cloud clustering on one side of the object surface, and the OAA module aligns the anchor points to the target position through the target-position-heatmap-guided attention operation. Finally, higher detection accuracy was achieved in sparse and crowded scenes. We compared the efficiency of multiple methods on the KITTI, JRDB, and blind-filling LiDAR datasets, and our method achieved the best results on each dataset.

## Figures and Tables

**Figure 1 sensors-23-08725-f001:**
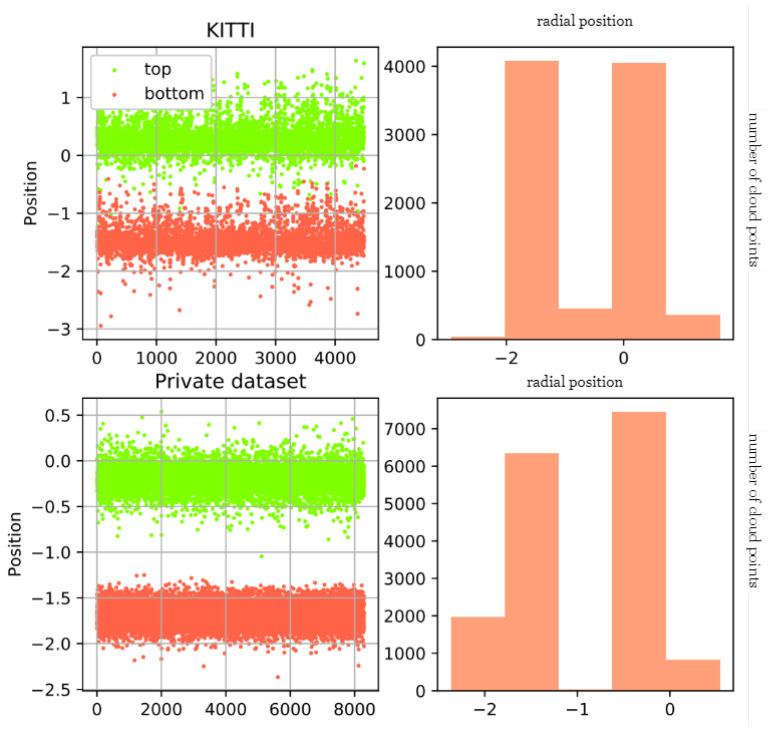
Dataset statistics results.

**Figure 2 sensors-23-08725-f002:**
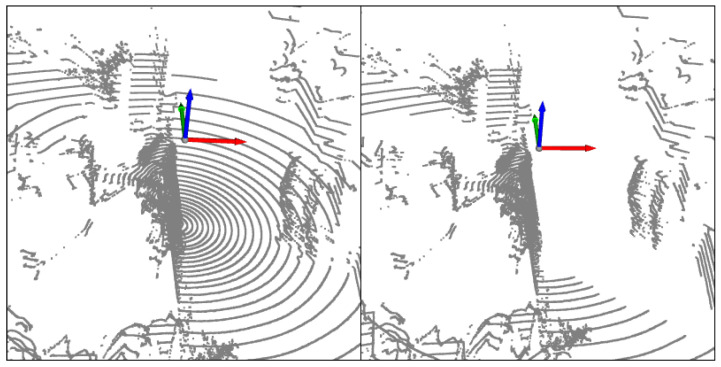
Filter point clouds based on Z-axis position.

**Figure 3 sensors-23-08725-f003:**
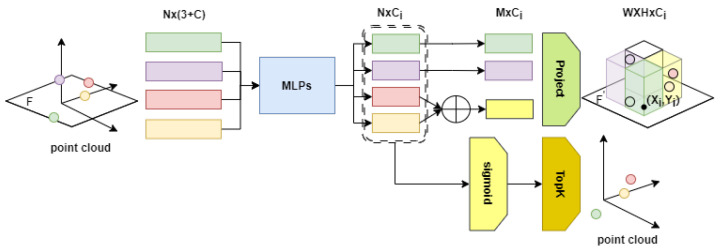
Attention-based point-resampling module.

**Figure 4 sensors-23-08725-f004:**
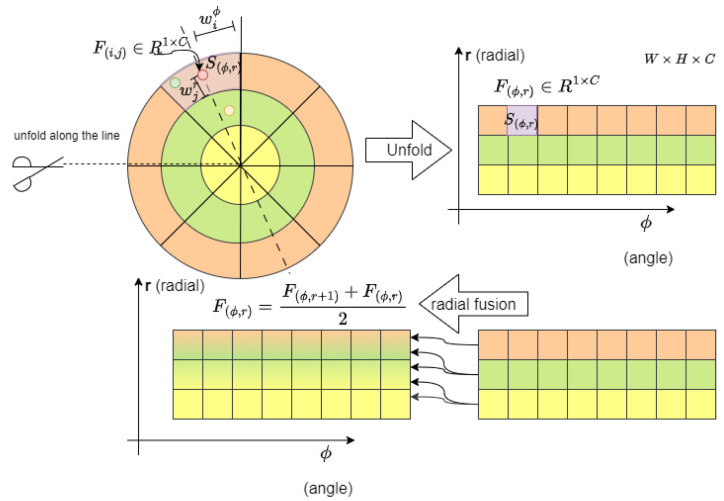
Center-alignment polar-quantization module.

**Figure 5 sensors-23-08725-f005:**
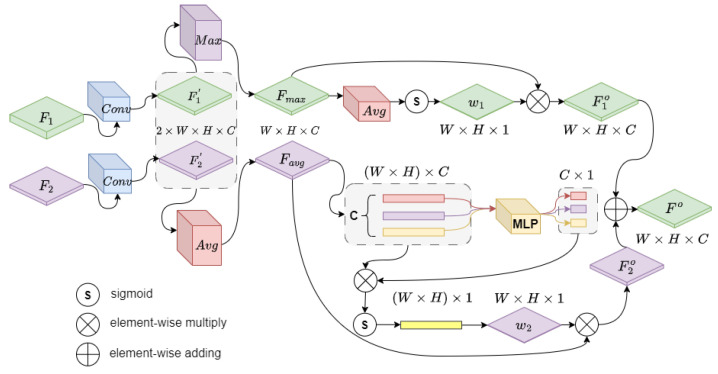
Object alignment attention module.

**Figure 6 sensors-23-08725-f006:**
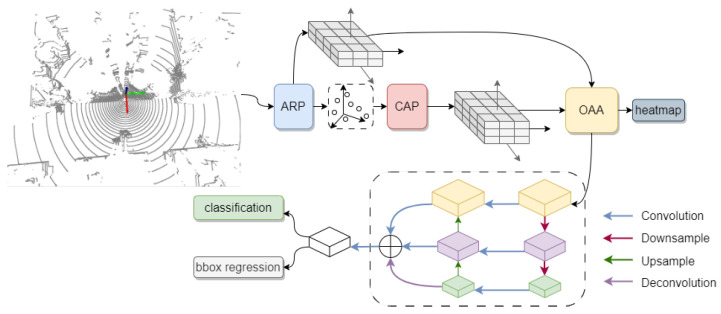
Detector framework.

**Figure 7 sensors-23-08725-f007:**
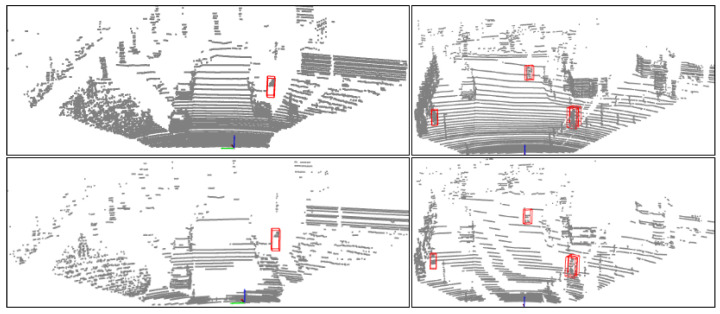
Low-resolution KITTI dataset.

**Figure 8 sensors-23-08725-f008:**
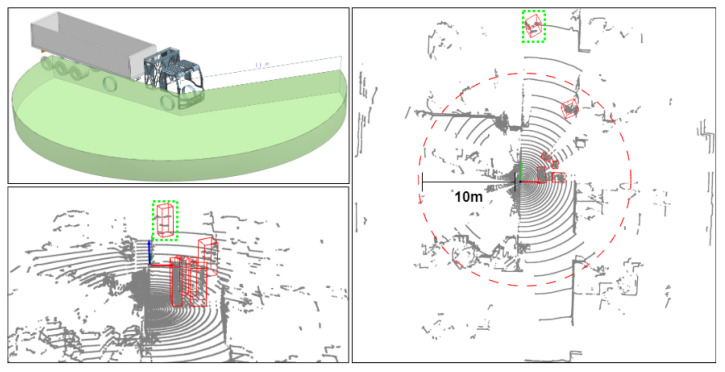
Blind-filling LiDAR dataset, the green dotted square means the labled objects in it are ignored.

**Figure 9 sensors-23-08725-f009:**
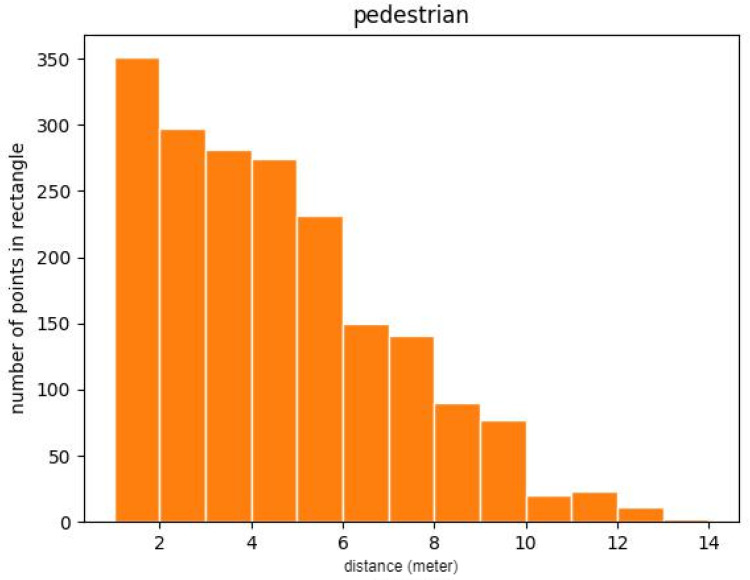
Distribution of the number of points based on the distance.

**Table 1 sensors-23-08725-t001:** Comparison of the effects of multiple network structures on the KITTI dataset.

Model	mAp(MeanAveragePrecision) ^1^	mAp ^2^	fps (Frames Per Second)
IA-SSD [10]	47.58	41.03	83
PointPillar [8]	47.81	41.60	88
3DSSD [11]	50.13	44.07	23
PiFeNet [22]	55.33	47.71	31
ours	55.54	47.85	54

^1^: The 64-line KITTI dataset; ^2^: z-axis-downsampled KITTI dataset.

**Table 2 sensors-23-08725-t002:** Comparison of effects of multiple network structures on the JRDB validation subset.

Model	AP@0.5 (Average Accuracy at 50% Intersection Ratio)	fps
PointPillar [8]	33.77	86
PiFeNet [22]	38.73	33
ours	39.11	55

**Table 3 sensors-23-08725-t003:** Comparison of effects of multiple network structures on the blind-filling datasets.

Model	mAp (IoU = 0.5:0.9 (Intersection Ratio from 50% to 90%))	fps
IA-SSD [10]	50.63	88
PointPillar [8]	50.01	94
3DSSD [11]	51.77	30
PiFeNet [22]	53.79	34
ours	53.92	52

**Table 4 sensors-23-08725-t004:** Results of ablation experiments.

${\mathrm{APR}}^{1}$	DFPS ^1^	CAP	Voxel	Heatmap	mAp ^2^	mAp ^3^
	✓		✓		47.01	49.02
	✓		✓	✓	46.80	48.91
	✓	✓			47.03	49.18
	✓	✓		✓	47.57	52.43
✓		✓			47.61	52.41
✓		✓		✓	47.85	53.92

^1^: The point cloud resampling module; ^2^: test on 32-line KITTI validation subset with IoU 0.5:0.9; ^3^: test on blind-filling LiDAR dataset validation subset with IoU 0.5:0.9.

## Data Availability

Data available upon request from the authors.

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
