# Peer review of "Real-Time 3D Object Detection on Crowded Pedestrians"

_sensors, 2023, doi:10.3390/s23218725_

Round 1

Reviewer 1 Report

1)     The paper language needs deep revision since it makes it difficult to grasp the idea at first. My suggestion for authors is to ask the native English speakers to assist with writing this paper in more simple way.
Here is just some few examples, but I would like to see the deep English revision throughout the paper. For examples:

In abstract:  indispensable à essential

Introduction: ``Point cloud of closer targets being easily confused’’? I think the authors means that clustering and associating cloud points to different pedestrian become difficult.

What is the rough means in this statement: ``receptive field of the model is relatively rough’’?

2)     The main contribution of the paper is in speed and accuracy that has been reported in Table 1-3. However, the citations of the former studies is missed. So, we cannot provide the fair judgement if we cannot see and refer properly to those works. Please add the citations for IA-SS, Point Pillar, etc.

3)     What is Distance measurements unit in Figure 9? Is it meter, centimeter or etc.?

4)     Why the blind filling data asset is private? What is the plan for sharing this dataset in the future? Can we have access and see the dataset samples?

5)     The abbreviated terms that have been used for results need to be expanded for the first time. IoU, mAp, fps, etc.

6) Explanation about the right hand plots of the Figure 1? What is the label for the x axis?

The paper language needs deep revision since it makes it difficult to grasp the idea at first. My suggestion for authors is to ask the native English speakers to assist with writing this paper in more simple way.
Here is just some few examples, but I would like to see the deep English revision throughout the paper. For examples:

In abstract:  indispensable à essential

Introduction: ``Point cloud of closer targets being easily confused’’? I think the authors means that clustering and associating cloud points to different pedestrian become difficult.

What is the rough means in this statement: ``receptive field of the model is relatively rough’’?

Author Response

Thanks for the advice.

1)I had done the deep English revision with help of  the native English speakers;

2) I had added the citations for IA-SSD、PointPillar,etc

3) I had added distance measurements unit in Figure9

4) I had provided an explanation for the data openness plan

5) I had expanded the abbreviated terms

6) I had added the labels of figure1

Reviewer 2 Report

Many current autonomous driving strategies include low-cost blind-filling LiDAR on the vehicle side to provide 360° full-area blind-free sensing. The target detection research program for high-density LiDAR, however, is inapplicable due to the significant sparsity of the blind-filling LiDAR. There are currently fewer studies for blind-filling LiDAR, and the approach proposed in this article can use blind-filling LiDAR to achieve improved detection accuracy and efficiency in sparse and congested circumstances, allowing blind-filling LiDAR to land. Overall, the paper is well-organized and its findings may provide some guidance for object detection. However, several minor problems must be addressed: 1 The introduction part of the article is not enough to introduce the background of the 3D detection of pedestrians, and it is suggested to supplement it. 2 There are a few wrong expressions in this paper, and it is recommended to carefully check and correct them. Such as page nine (Our model can achieve a running speed of 50fps), however the value in table2 and table3 is over 50. page nine (…as showed in the 3D box marked in green in Figure 8, we directly ignore it.) should probably be changed to (…in red in Figure 8, we directly ignore it.)

3 On the page 2, the fourth point may not be the contribution point, this just shows the effect of the proposed model is verified.

4 All symbols in the formula should be explained, such as formulas (1) and formulas (2).

The English proficiency of the manuscript needs further improvement.

Author Response

Thanks for your adivice.

1) I've added more on the background and development of pedestrian detection under 3D point clouds.

2) I've corrected some of the text that was expressed incorrectly.

3)I have deleted point 4 which is not a contribution point.

4)I've added a description of the symbols in the formulas.

Round 2

Reviewer 1 Report

Thanks to the authors for addressing my comments. I have no more questions.

The paper has been revised, some minor English revisions might be required.